# Unique Interaction between Layered Black Phosphorus and Nitrogen Dioxide

**DOI:** 10.3390/nano12122011

**Published:** 2022-06-10

**Authors:** Jingjing Zhao, Xuejiao Zhang, Qing Zhao, Xue-Feng Yu, Siyu Zhang, Baoshan Xing

**Affiliations:** 1Key Laboratory of Pollution Ecology and Environmental Engineering, Institute of Applied Ecology, Chinese Academy of Sciences, Shenyang 110016, China; zjj911051836@126.com (J.Z.); zhangxuejiao@iae.ac.cn (X.Z.); zhaoqing@iae.ac.cn (Q.Z.); 2Shenyang Institute of Applied Ecology, University of Chinese Academy of Sciences, Beijing 100049, China; 3Key Laboratory of Integrated Agro-Environmental Pollution Control and Management, Institute of Eco-Environmental and Soil Sciences, Guangdong Academy of Sciences, Guangzhou 510650, China; 4China National-Regional Joint Engineering Research Center for Soil Pollution Control and Remediation in South China, Guangzhou 510650, China; 5Materials and Interfaces Center, Shenzhen Institutes of Advanced Technology, Chinese Academy of Sciences, Shenzhen 518055, China; xf.yu@siat.ac.cn; 6Stockbridge School of Agriculture, University of Massachusetts, Amherst, MA 01003, USA; bx@umass.edu

**Keywords:** layered black phosphorus, vacancy defect, single electron, nitrogen dioxide, hazardous gas pollutants

## Abstract

Air pollution caused by acid gases (NO_2_, SO_2_) or greenhouse gases (CO_2_) is an urgent environmental problem. Two-dimensional nanomaterials exhibit exciting application potential in air pollution control, among which layered black phosphorus (LBP) has superior performance and is environmentally friendly. However, the current interaction mechanism of LBP with hazardous gases is contradictory to experimental observations, largely impeding development of LBP-based air pollution control nanotechnologies. Here, interaction mechanisms between LBP and hazardous gases are unveiled based on density functional theory and experiments. Results show that NO_2_ is different from other gases, as it can react with unsaturated defects of LBP, resulting in oxidation of LBP and reduction of NO_2_. Computational results indicate that the redox is initiated by p orbital hybridization between one oxygen atom of NO_2_ and the phosphorus atom carrying a dangling single electron in a defect’s center. For NO, the interaction mechanism is chemisorption on unsaturated LBP defects, whereas for SO_2_, NH_3_, CO_2_ or CO, the interaction is dominated by van der Waals forces (57–82% of the total interaction). Experiments confirmed that NO_2_ can oxidize LBP, yet other gases such as CO_2_ cannot. This study provides mechanistic understanding in advance for developing novel nanotechnologies for selectively monitoring or treating gas pollutants containing NO_2_.

## 1. Introduction

Hazardous gases emitted from industries, traffic and other causes cause a series of climate, environmental and health problems, including global warming, acid rain and photochemical smog [1,2,3]. According to International Energy Agency’s report in 2016, 6.5 million premature deaths can be attributed to air pollution worldwide annually [4]. Exposure during pregnancy and early postnatal periods to nitrogen dioxide (NO_2_) or sulfur dioxide (SO_2_) is associated with childhood allergy diseases [5]. Adsorbents are increasingly used in eliminating hazardous gases, such as molecular sieves [6], porous carbon [7], metallic clusters [8], graphene [9] and MXenes [10]. Two-dimensional (2D) nanomaterials have great advantages in gas adsorption over other materials, owing to quantum size effects, large surface areas, excellent electronic/photoelectronic performance, etc. [11,12]. A comprehensive understanding of the interaction mechanisms between nanomaterials and hazardous gases would be beneficial to developing efficient and selective nanotechnology for prevention and control of air pollution.

Layered black phosphorus (LBP) is a newly popular 2D nanomaterial [13]. Its large specific surface area (2400 m^2^/g [14]) offers plenty of interaction sites for hazardous gas molecules. Recent studies indicate LBP is a nanomaterial superior to graphene or molybdenum disulfide as a gas sensor for detecting a variety of hazardous gases [15,16]. LBP is attractive due to inherent biocompatibility [17], and was found to be nontoxic to L-929 fibroblasts at concentrations lower than 4 μg mL^−1^ [18]. As a degradable nanomaterial, the decomposition of LBP mainly produces low-toxicity phosphate, phosphite, etc. [19] Therefore, LBP has a controllable environmental risk and is suitable for the development of sustainable nanotechnology for monitoring or eliminating hazardous gas pollutants. It is necessary to understand the interaction mechanism of hazardous gases with LBP.

Compared to other gasses LBP has the highest sensitivity, superior selectivity and a short response time toward NO_2_ [20,21]. As low as 5 ppb of NO_2_ can induce a clear conductance change in an LBP sensor [16]. A similar response cannot be obtained for carbon monoxide (CO), carbon dioxide (CO_2_) or ammonia (NH_3_) until the concentrations increase to 1 × 10^2^, 1 × 10^4^ and 10 ppm, respectively [22]. An LBP sensor demonstrated a low limit of detection of 0.4 ppb toward NO_2_ under N_2_ and air conditions [23]. Recent studies show that incomplete recovery of an LBP sensor is remarkable after exposure to 50 ppb NO_2_ [24,25]. The high sensitivity/selectivity and poor reversibility of LBP sensors imply that the interaction of LBP with NO_2_ is undoubtedly stronger than those with other gases.

The adsorption energies (*E*_ad_) of NO_2_ (−0.41 to −0.27 eV [15,26]) and nitrogen monoxide (NO) (−0.26 to −0.18 eV [26,27]) on a perfect LBP are fairly similar to those of other hazardous gases, including CO_2_, CO and SO_2_ (−0.55 to –0.12 eV [28,29,30]), based on the generalized gradient approximation with the parameterization of Perdew−Burke−Ernzerhof (GGA-PBE) computation methods. As for favorable adsorption configurations, no consensus has been achieved by different computational studies for NO_2_ [15,31] and for SO_2_ [32,33,34]. Accordingly, computational studies so far suggest that NO_2_ is not remarkably different from other gases in terms of interacting with LBP.

The inconsistency between experimental results and theoretical computations impedes the development of LBP-based nanotechnology in the field of air pollution control. To break through the knowledge barrier, understanding the interaction mechanisms of LBP with various hazardous gases is necessary. First, NO_2_ or NO has an odd number of valence electrons [35,36] (Appendix A), and therefore is inferred to act more actively while interacting with nanomaterials than gas pollutants owning even numbers of valence electrons [37,38]. In addition, various defects, including Stone–Wales (SW), and single and double vacancy (SV and DV), are inevitably formed during the fabrication of LBP [39]. Due to loss of neighboring atoms, atoms in the vacancy are unsaturated, and easily form free radicals carrying single electrons [40]. Previous studies show that SV defects of LBP significantly improve adsorption of phosphine and arsine [41]. Vacancy defects can significantly promote binding energies between blue phosphorene and volatile organic compounds [42]. Accordingly, we guess that unsaturated defects carrying dangling single electrons in LBP are responsible for the distinct interaction with NO_2_.

In this study, the interaction mechanisms of LBP and common hazardous gases are unveiled based on density functional theoretical (DFT) computations and experiments. NO_2_, NO, NH_3_, SO_2_, CO_2_ and CO were selected as representative hazardous gas pollutants, as they possess different outer sphere electronic structures (Appendix A). Two hypotheses, associated with the electronic structures of gas molecules and defective properties of LBP, respectively, were tested to probe the interaction mechanism between NO_2_ and LBP. The results provide new insights on the interaction mechanisms of LBP with hazardous gas pollutants and are beneficial to developing sustainable nanomaterials for air pollution prevention and control. 

## 2. Computational Methods

### 2.1. Adsorbent Model

A 3 × 1 × 3 supercell containing 36 phosphorus atoms (P_1_–P_36_) was constructed to simulate perfect SLBP. Each P atom of SLBP is covalently bonded to three adjoining P atoms, forming a wrinkled honeycomb structure (Appendix A). This wrinkled honeycomb structure lets SLBP contain two atomic layers, each including 50% of the atoms. The size of the periodic box was *a* = 9.845 Å, *b* = 17.097 Å and *c* = 13.972 Å along zigzag, vertical and armchair directions, respectively. The vacuum region in the vertical (*b*) direction was set to be 15 Å to avoid interactions of gas molecule with SLBP in the adjacent periodic box. Similar box size was used for simulating the LBP-based gas sensor [31]. Defective SLBP (d-SLBP) was constructed on the basis of optimized geometry of SLBP. The initial configurations of SLBP and d-SLBP are shown in Appendix A. Additional information is included in the Appendix A. Two in-plane SW1 and SW2 [39,43] defects were built by rotating a vertical bond P_18_–P_20_ or a horizontal bond P_17_–P_18_ by 90°. SV and DV1-DV3 defects were built by deleting phosphorus atoms. Theoretically, deleting one phosphorus atom creates three neighboring unsaturated phosphorus atoms, each carrying one dangling unpaired electron. SV was constructed by removing atom P_18_. DV1 and DV2 were built by removing two bonded phosphorus atoms, P_18_–P_20_ and P_17_–P_18_, respectively. DV3 was obtained by removing two separate atoms, P_18_ and P_29_. Edge defects in either zigzag (EGz1 and EGz2) or armchair (EGa1 and EGa2) direction were constructed, considering the anisotropy of LBP [44]. The periodic box was enlarged in the armchair (*c*) or the zigzag (*a*) direction to 20 Å for creating edge defects. Box enlargement inevitably results in unsaturated phosphorus atoms on edges. By saturating phosphorus atoms on one edge with hydrogen atoms, EGz1 and EGa1 were built. By deleting one phosphorus atom from EGz1 (P_31_) or EGa1 (P_21_), EGz2 or EGa2 was constructed. Fully optimized geometries of SLBP and d-SLBP are shown in Figure 1. 

### 2.2. Adsorption Complex Models

Generally, there are three widely accepted adsorption sites in LBP, including top, bridge and hollow types [29]. In this study, the three adsorption sites were further differentiated between SLBP and d-SLBP to be three top (T, T1, T2), 4 bridge (B, B1–B3) and 5 hollow (H, H1–H4) sites (Figure 1). Gas molecules were placed at one of these sites as initials. Due to the influences of spatial structures, the adsorption site for gas molecules in the fully relaxed configuration of the adsorption complex was classified according to the closest type. Different initial distances (ca. 2.14 to 4.15 Å) between gas molecule and SLBP or d-SLBP were tested in the computation. No direct bindings between gas molecules and any phosphorus atoms existed in initial configurations.

Orientations of gas molecules were comprehensively considered in the computations, as previous computational results for orientations of gas molecules adsorbed on LBP were obscure [15,31,32,33,34,39]. For nonlinear molecules (NO_2_, SO_2_, NH_3_), vertical orientations, including single atom pointing (i.e., **1O*v***, **N*v***, **S*v***, **1H*v***) or multiple atoms pointing (**2O*v***, **3H*v***) to SLBP plane, and parallel orientations (***p***), were included. For linear molecules (CO_2_, NO, CO), vertical single atom pointing (**O*v*, C*v*, N*v***) and parallel (***p***) orientations were calculated. Considering the anisotropy of LBP, orientations were further differentiated to be along zigzag (***z***) or armchair (***a***) directions. Taking **2O*vz*** as an example, it represents a configuration where two oxygen atoms of a dioxide point to the SLBP plane, and the orientation of the dioxide is along the zigzag direction. The initial configurations of gases with different orientations on SLBP are shown in Appendix A.

### 2.3. Computational Methods

All computations were performed with a CASTEP program in the Material Studio (Biovia, San Diego, CA) software [45]. The GGA-PBE [46] rather than LDA (Local Density Approximation) [47] was used to describe exchange-correlation functional, as LDA was regarded to usually underestimate bonding distance and overestimate binding energy. In addition, van der Waals (vdW) correction was performed within the empirical correction scheme of Grimme [48] method. The Brillouin zone was sampled using a 4 × 1 × 3 Monkhorst-Pack *k*-point grid. The Kinetic energy cutoff of 500 eV was used in geometry optimization process. Convergence tests for energy and *k*-point are shown in Appendix A. Initial configurations were fully optimized until the force on each atom was less than 0.05 eV/Å and the energy tolerance was less than 1 × 10^−5^ eV/atom. Spin polarization was included for computing NO, NO_2_ or unsaturated d-SLBP (SV, DV1, DV3 and edge defects). Band structure and Mulliken population were analyzed based on fully optimized geometries. A large *k*-point (10 × 1 × 8) was used to achieve high accuracy in density of states (DOS) computations. The calculation methods of adsorption energies including (*E*_ad_) and not including (*E*_0_) vdW interaction energy (*E*_vdW_) corrections and deformation energies (*E*_def_) of d-SLBP after adsorbing gas molecules are described in Appendix A.

### 2.4. Experimental Methods

LBP was prepared by the liquid exfoliation method described in a previous study [19]. Firstly, the bulk black phosphorus was ground into powders with an agate mortar, and transferred into oxygen-free Millipore ultrapure water in the glovebox. Then, the powders were sonicated with a probe for 12 h. The LBP suspension was transferred to centrifuge tubes in a glovebox, and then centrifuged at 10,000 rpm for 30 min. The supernatant was transferred to the glovebox and filtered through a 0.22 μm cellulose membrane to collect LBP. After drying overnight in the glovebox, the LBP was transferred into three anaerobic bottles. Two of the bottles were continuously pumped with NO_2_ (200 ppm NO_2_ in N_2_, Beijing Lvyuan Dade Biological Tech Co., Ltd., Beijing, China) and CO_2_ (>99.5%, Shenyang Shuntai Special Gas Co., Ltd., Shenyang, China) at a flow rate of 10 mL/min for 3 h. Then, the two bottles were sealed, allowing thorough interaction of LBP with the gases. After 24 h, each of the two bottles was injected with pure N_2_ for 1 h to remove excess NO_2_ or CO_2_. LBP in the third bottle was filled with pure N_2_ (>99.99%, Shenyang Shuntai Special Gas Co., Ltd.) for 3 h and placed in the glovebox for over 24 h. X-ray photoelectron spectroscopy (XPS, ESCALAB250, Thermo VG, Waltham, MA, USA) was used to identify phosphorus components of LBP after treatment with N_2_, NO_2_ or CO_2_.

## 3. Results and Discussion

### 3.1. Adsorption of Gas Molecules on Perfect LBP

A perfect single-layer LBP (SLBP) contained in a periodic box was used in computations (Figure 1a). Initial configurations of adsorption complexes were constructed by varying orientation, adsorption sites or distances of gas molecules on the plane of SLBP, in order to solve the inconsistencies in *E*_ad_ values and in favorable adsorption configurations among previous computational studies. By comprehensively screening hundreds of adsorption configurations, the most stable adsorption configuration (adsorption complexes possessing the most negative *E*_ad_ values) of each gas molecule was obtained (Appendix A). Computational results indicate that orientations, together with distances of gas molecule toward the LBP plane, in the initially constructed adsorption configurations were key factors influencing the computational results on favorable adsorption configurations and *E*_ad_ (Appendix A). These two factors were the primary reasons for contrary computational results in previous studies [15,31,32,33,34,37] (Appendix A). In comparison with initial orientations of gas molecules, impacts of initial adsorption sites on favorable configurations or *E*_ad_ are not remarkable.

According to *E*_ad_ values (Table 1), the two oxygen atoms in pointing adsorption configuration (*E*_ad_ = −0.225 eV, Appendix A) are more stable than the nitrogen atom in pointing configuration (*E*_ad_ = −0.211, −0.202 eV, (Appendix A) in NO_2_. This is different from the adsorption of NO_2_ on other 2D nanomaterials, such as graphene or indium nitride, where NO_2_ is bound to the sheet’s surface with the nitrogen end [49]. In the case of SO_2_, two oxygen atoms in pointing configuration are less favorable (Appendix A). Parallel and sulfur atom pointing configurations showing comparatively the lowest *E*_ad_ values (−0.310, −0.309 eV) are the most favorable adsorption configurations for SO_2_ (Appendix A). Three linear molecules, CO_2_, NO and CO, prefer parallel configurations, as indicated by *E*_ad_ values. In the most stable adsorption configurations, CO_2_ is almost completely parallel to the SLBP plane with a negligible dihedral angle (Appendix A), whereas for NO or CO, the dihedral angle to the SLBP plane is 9–17° (Appendix A). This small dihedral angle is mainly attributed to the asymmetry of monoxide molecules. Differently from NO_2_, the nitrogen atom pointing configuration is the most stable configuration for NH_3_ (Appendix A).

The small *E*_ad_ values (−0.310 to –0.138 eV), low charge transfer amounts (−0.15 to 0.04 e) and large distances to the SLBP plane (2.47–3.44 Å) indicate weak interaction between gas molecules and SLBP (Table 1). The distance of the N atom in NO to SLBP (2.47 Å) is the shortest in the most stable adsorption configurations of the six gas molecules, but is still much longer than a P–N covalent bond length (1.89 Å [50,51]). The adsorption of gas molecules on perfect SLBP plane is dominated by vdW forces, as indicated by the ratio of 57–82% relative to *E*_ad_.

Based on the first guess, adsorption of NO_2_ and NO carrying odd numbers of valence electrons on LBP should be stronger than that of other gas molecules. However, the *E*_ad_ value of SO_2_ is the largest among all gas molecules. The computed *E*_ad_ value of NH_3_ is between that of NO and NO_2_. This is not in accordance with the first guess or experimental results, which indicate that the sensitivity of LBP sensors toward NH_3_ (10 ppm) is about 3–5 orders of magnitude lower than that toward NO_2_ (0.4–20 ppb) in a dry environment [16,22,23]. Accordingly, there must be other mechanisms responsible to the distinct interaction between LBP and NO_2_.

### 3.2. Adsorption of Gas Molecules on Defects in LBP

A variety of in-plane and edge defective SLBP (d-SLBP) models were built. Optimized geometries of d-SLBP and computed formation energies can be found in Figure 1b–k and Appendix A. Dioxides (NO_2_, SO_2_ or CO_2_) were selected in the computations to investigate the role of the outmost valence shell of gases while interacting with d-SLBP.

*E*_ad_ values computed for adsorption of dioxides on all defective adsorption sites of each d-SLBP are shown in Figure 2a–c. By comparing *E*_ad_ values of the most stable adsorption configurations of NO_2_ on each d-SLBP (Figure 2d and Appendix A), we found that four among ten d-SLBP, including SV, DV3, EGz1 and EGa2, exhibit large *E*_ad_ values (−2.584 to –1.304 eV). This means that strong chemisorption is formed. In the most stable adsorption configurations of the four d-SLBP, a new P–O bond, 1.56–1.64 Å in length, is formed between NO_2_ and a phosphorus atom in a defect center (Figure 2e). This distance is close to or falls within the range of length of P–O single bond (1.593 [52]–1.75 [53] Å) suggested for LBP. As a result, one O atom is pulled out from NO_2_, leaving a part containing one N atom and one O atom (**[NO]**). To know the nature of **[NO]**, it was pulled away from the main part of d-SLBP, taking SV as an example to construct a desorption system by preventing any interaction between the two parts. Total charge and total spin population distributed on **[NO]** were 0 and 1 (Appendix A), respectively. In other words, NO was formed during the interaction of the 4 d-SLBP and NO_2_. After separating from **[NO]**, the newly formed P–O in d-SLBP was shortened to 1.52 Å, which is close to the length of a P=O double bond (1.445–1.502 Å [52]).

Interestingly, in comparison with SV, EGz1 or EGa2 (*E*_ad_ = −1.543 to –1.304 eV), DV3 possess a larger *E*_ad_ value (−2.584 eV), not only interacting with NO_2_, but also with SO_2_ or CO_2_ (*E*_ad_ = −1.378 to –1.211 eV), as shown in Figure 2d. However, no new bonds are formed between DV3 and SO_2_ or CO_2_ (*d*_shortest_ = 2.89–3.66 Å), indicating the negative *E*_ad_ values are unusually large. This is because non-negligible deformation happens to DV3 while interacting with dioxides, resulting in inauthentic interaction energies. To obtain real interaction energies between dioxides and d-SLBP, deformation energies (*E*_def_) were computed for each d-SLBP (Appendix A). Taking NO_2_ as an example, the *E*_def_ values of DV3 (−0.718 eV) are higher than those of other d-SLBP (−0.013 to −0.254 eV), implying that the deformation of other d-SLBP is slight. After deducting *E*_def_ from *E*_ad_, the modified interaction energy (*E*_ad-def_) of NO_2_ and DV3 is −1.866 eV, which is comparable to those of SV, EGz1 and EGa2 (−1.543 to –1.304 eV) (Appendix A). Accordingly, the four d-SLBP show similar interaction strengths with NO_2_ and significantly improve the adsorption of NO_2_ compared with SLBP. The *E*_ad-def_ values of SO_2_ and CO_2_ interacting with DV3 are −0.304 and −0.137 eV, respectively, comparable to those computed for interacting with SLBP (Appendix A). Therefore, d-SLBP enhances only the interaction of LBP with NO_2_, but not with SO_2_ or CO_2_.

### 3.3. Interaction Mechanisms of Defective LBP and Gas Molecules

To probe interaction mechanisms between NO_2_ and d-SLBP, spin density distributions of d-SLBP were mapped (Figure 3). Spin density can explain the distribution of unpaired electrons or single electrons in space [54]. The results show that SV, DV3, EGz1 and EGa2 possess non-zero spin density, meaning that each of the four d-SLBP has unsaturated phosphorus atoms carrying unpaired dangling single electrons. This is caused by losing one or more neighbor P atoms. Accordingly, the unpaired electron of d-SLBP is inferred to be closely related to the unique interaction with NO_2_.

Taking SV as an example, there were three unsaturated atoms, **P**_13_, **P**_17_ and **P**_20_, in the initially constructed geometry due to loss of **P**_18_. After geometry optimization, a new bond **P**_13_–**P**_20_, 2.37 Å in length was formed, leaving one dangling atom, **P**_17_, in SV (Figure 1d). Most of the spin density (0.64) is distributed around atom **P**_17_ in SV (Figure 3c), indicating the dangling single electron mainly presents around atom **P**_17_. According to *E*_ad_ values, the most stable adsorption configuration of NO_2_ on the top of SV is that in which NO_2_ binds with atom **P**_17_ (Figure 2e). For the other three d-SLBP, the most stable adsorption configuration of NO_2_ also uses the phosphorus atoms carrying the highest non-zero spin density (Appendix A). Unsaturated carbons in defected graphene were found to be active toward NO_2_ molecules [55]. Therefore, the results indicate that the unique interaction with NO_2_ is attributable to unpaired single electron carried by the four d-SLBP.

Although interacting with d-SLBP carrying dangling single electrons, the adsorption of SO_2_ or CO_2_ is not enhanced, as indicated by the *E*_ad-def_ values (Appendix A). This means that the electron properties of gas molecules are another key factor governing interaction strengths between LBP and gases. To further prove this inference, interactions of the four unsaturated d-SLBP with NO and NH_3_ were computed. Similarly to NO_2_, NO has an odd number of outmost valence electrons. NH_3_ has an even number of outmost valence electrons, and every two of them are paired.

The *E*_ad-def_ values (Figure 4a) between NO and SV, DV3 or EGa2 are almost as large as that of NO_2_. The *E*_ad-def_ values between NO and EGz1 are a little lower, but still more negative than that between NO and SLBP. In the most stable adsorption configurations of the adsorption complex of each d-SLBP, the N atom of NO is near d-SLBP at a distance of 1.78–1.98 Å (Figure 4b). This distance is close to (in the case of SV, DV3, EGa2) or slightly larger (in the case of EGz1) than the length of a typical P–N single bond (commonly accepted value is 1.89 Å [50,51]), but much smaller than the distance (2.47 Å) between NO and SLBP (Table 1), which implies that there is a strong chemisorption between NO and unsaturated d-SLBP. The difference is that d-SLBP binds NO by forming a weak P–N bond but reacts with NO_2_ by abstracting one O atom in redox reactions. The original bond of NO (1.16 Å) was weakened but not broken (1.20 Å) by d-SLBP, but one N=O bond of NO_2_ was broken due to reaction with d-SLBP. The *E*_ad-def_ values or distances of NH_3_ interacting with unsaturated d-SLBP are comparable to those of NH_3_ interacting with SLBP (Figure 4a and Appendix A). This confirms that the outmost valence electron characteristics of gas molecules are responsible for strong interactions with LBP.

The above computational results show that NO_2_ can oxidize unsaturated LBP defects to produce oxidized phosphorus and NO. This redox mechanism has never been reported for NO_2_ and other two-dimensional nanomaterials as far as we know. Previous studies showed that vacancy defects in graphene [56] or tungsten trioxide [57] can improve the adsorption of NO_2_, but did not explain the reason. Yan et al. [42] showed that unsaturated carbon atoms in carbon vacancy defects of graphene can bind the nitrogen atom of NO_2_. Carbon vacancies produced chemisorption sites on graphene for NO_2_ [58]. Therefore, the enhanced adsorption of NO_2_ on graphene is most likely due to strong chemisorption similar to that of NO and unsaturated LBP defects, but not a redox reaction.

### 3.4. Orbital Analysis on the Nature of Interaction

Total density of states (TDOS) and partial density of states (PDOS) were computed to further understand interaction mechanism of NO_2_ with LBP (Appendix A). Taking SLBP and SW1 as examples of saturated LBP, new impurity states that emerged in conduction bands (CB) of the adsorption complexes at around 2 eV were mainly contributed by *p* orbitals of O and N atoms, and those in valence bands (VB) at −4 to –2 eV were contributed by *p* orbitals of O atoms (Figure 5a,b). Slight *p* orbital hybridization of O, N and P atoms near the Fermi level can explain the small amount of charge transfer (−0.11 e, −0.14 e) between NO_2_ and neighboring P atoms (Table 1). Since there is no single electron that can interact with these NO_2_ in SLBP and SW1, the whole system exhibits spin asymmetry, which is mainly derived from the nitrogen and oxygen of NO_2_.

Taking SV as an example of unsaturated LBP, an individual SV owns an odd number of single electrons, as shown by asymmetric TDOS (Appendix A). As NO_2_ carries an odd number of valence electrons, the total number of valence electrons of adsorption complexes formed by SV and NO_2_ is even. However, the results show that the complex has a spin asymmetric TDOS (Figure 5c), indicating that electrons are unpaired. This is a cue for the presence of single-electron species in the complex of NO_2_ and SV, i.e., [NO] (a NO molecule) carrying an unpaired valence electron. Significant orbital overlaps indicting strong hybridization between p orbitals of O and **P**_17_ was observed, indicating formation of a P–O bond. Similar results were observed for other d-SLBP; see Appendix A for detailed discussion.

Large and continuous orbital overlaps were also observed for adsorption complexes of NO and unsaturated d-SLBP (Appendix A), but not for those with SO_2_ or CO_2_ (Text S9). This explains how NO can form strong chemisorption with d-SLBP, but SO_2_ or CO_2_ cannot. The orbital overlap of NO with in-plane vacancy defects (SV and DV3) is stronger than that with edge defects (EGa2 and EGz1), in accordance with the order of the *E*_ad_ values (DV3 > SV > EGa2 > EGz1). Differently from NO_2_, TDOS of the adsorption complex of NO and SV is spin symmetric (Appendix A), indicating that all electrons are paired. As both individual NO and individual SV have unpaired electrons, the spin symmetry of adsorption complex is a clear clue for the formation of a P–N=O moiety (Appendix A).

### 3.5. Experimental Verification of Oxidation of LBP by NO_2_

According to the computational results, the unique interaction between NO_2_ and LBP is redox in nature, as the unsaturated defect in LBP is oxidized and NO_2_ is reduced. To verify that LBP interacts with NO_2_ differently from other gas pollutants, experiments were performed. LBP was prepared following the method described in Appendix A. As defects are inevitably formed during the exfoliation of LBP in water, the as-prepared LBP was exposed to NO_2_ in order to check whether oxidation can occur. As a comparison, the as-prepared LBP was exposed to CO_2_, which has no unpaired valence electrons and interacts with LBP mainly through van der Waals interactions based on our computational results. LBP exposed to N_2_ was used as a blank control to show the oxidation status of the original LBP.

After 24 h of exposure, LBP nanoflakes were characterized by X-ray photoelectron spectroscopy. Characteristic peaks corresponding to P 2p1/2 and P 2p3/2 [59] were observed at 130.4 and 129.6 eV. No oxidation peaks were found when BP was exposed to N_2_ (Figure 6a) or CO_2_ (Figure 6b). However, after exposure to NO_2_, a noticeable peak emerged slightly above 133 eV (Figure 6c), corresponding to oxidized phosphorus (PO_x_) [60]. The PO_x_ peak around 133 eV accounts for 23.3% of total phosphorus contents in LBP. This confirms that LBP can be rapidly oxidized by NO_2_. The PO_x_ peak (if there is any) in Figure 6b is as negligible at that in Figure 6a, indicating that direct interaction between oxygen atoms of CO_2_ with LBP is negligible. These experimental results give solid evidence supporting the computational conclusion that NO_2_ can oxidize LBP, whereas CO_2_, being without an unpaired single electron, cannot.

## 4. Conclusions

In summary, an interaction mechanism of LBP with common gas pollutants was unveiled in this study. LBP can react with NO_2_ following a redox mechanism, which is essentially different from the physisorption of SO_2_, NH_3_, CO_2_ and CO, or chemisorption of NO. The adsorption mechanism of gas molecules on LBP was first clarified. The unique interaction with NO_2_ is owed to unsaturated phosphorus carrying a single electron in LBP. This mechanism not only solves the problems puzzling experimental studies on LBP-based gas sensors, but also provides ideas for the development of environmentally friendly nanotechnology based on LBP to monitor or treat hazardous gas pollutants.

## Figures and Tables

**Figure 1 nanomaterials-12-02011-f001:**
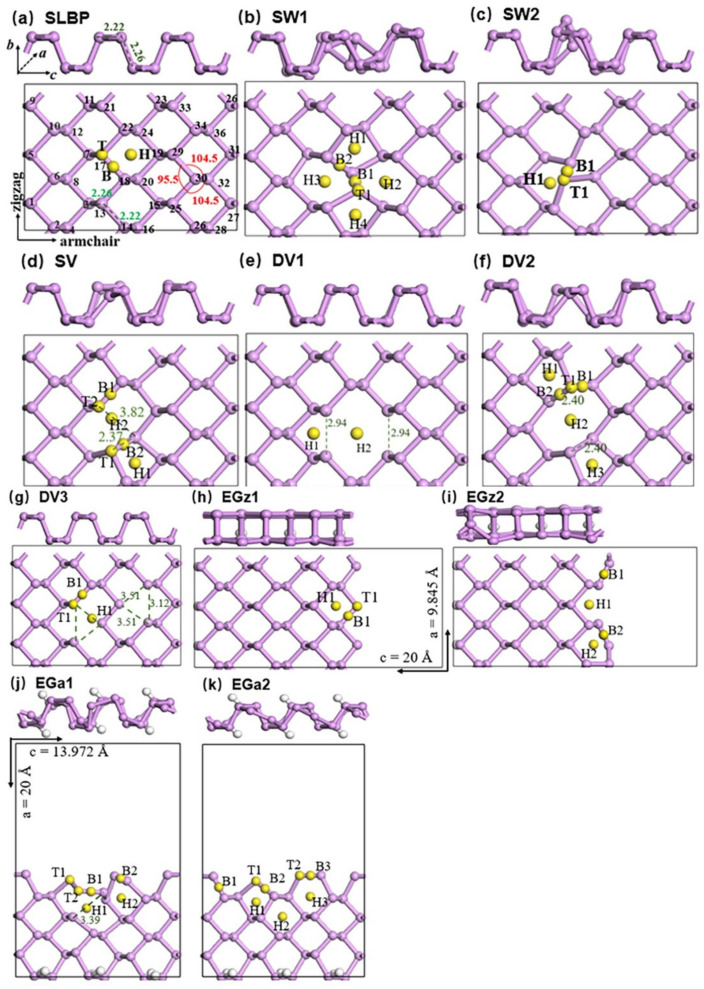
Side and top views of fully optimized geometries of SLBP and d-SLBP: (**a**) SLBP, (**b**) SW1, (**c**) SW2, (**d**) SV, (**e**) DV1, (**f**) DV2, (**g**) DV3, (**h**) EGz1, (**i**) EGz2, (**j**) EGa1 and (**k**) EGa2. T, T1, T2: top sites; B, B1–B3: bridge sites; H, H1–H4: hollow sites.

**Figure 2 nanomaterials-12-02011-f002:**
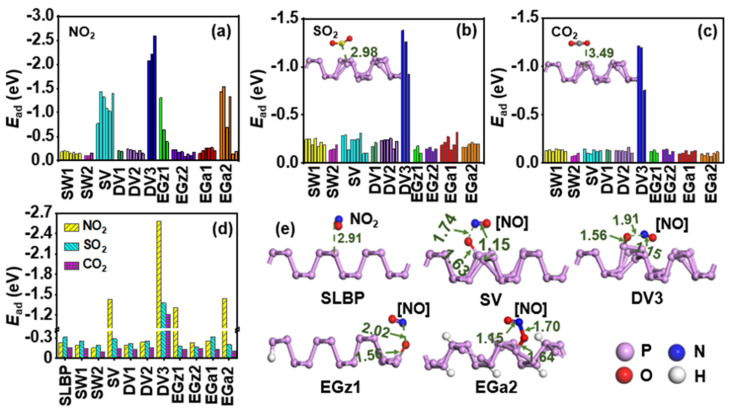
Adsorption energies (*E*_ad_, eV) computed for stable adsorption configurations of NO_2_ (**a**), SO_2_ (**b**) or CO_2_ (**c**) on different adsorption sites of d−SLBP. *E*_ad_ values of the most stable adsorption configurations (**d**). The most stable adsorption configurations of NO_2_ on SLBP, SV, DV3, EGz1 and EGa2 (**e**).

**Figure 3 nanomaterials-12-02011-f003:**
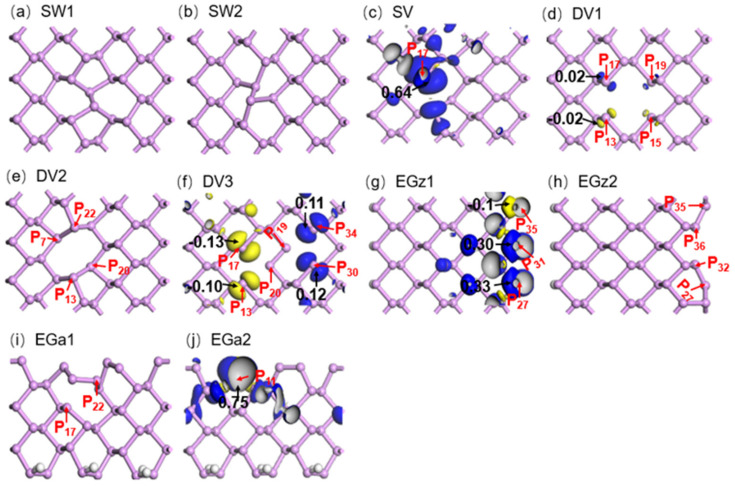
Spin density distribution on d−SLBP. (**a**) SW1, (**b**) SW2, (**c**) SV, (**d**) DV1, (**e**) DV2, (**f**) DV3, (**g**) EGz1, (**h**) EGz2, (**i**) EGa1 and (**j**) EGa2. Blue and yellow isosurfaces mean non−zero positive and negative spin density, respectively. Black numbers: spin population. Red numbers: Labels of phosphorus atoms.

**Figure 4 nanomaterials-12-02011-f004:**
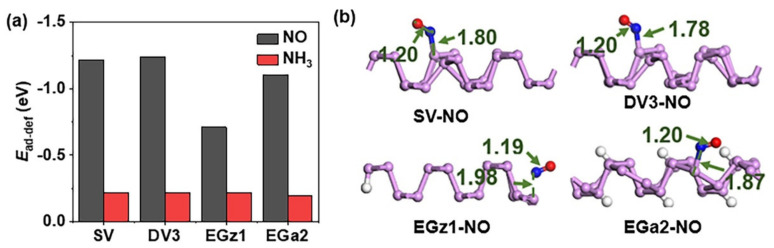
(**a**) Modified interaction energy (*E*_ad−def_, eV) for adsorption of NO and NH_3_ on SV, DV3, EGz1 and EGa2. (**b**) The most stable adsorption configurations of NO on SV, DV3, EGz1 and EGa2.

**Figure 5 nanomaterials-12-02011-f005:**
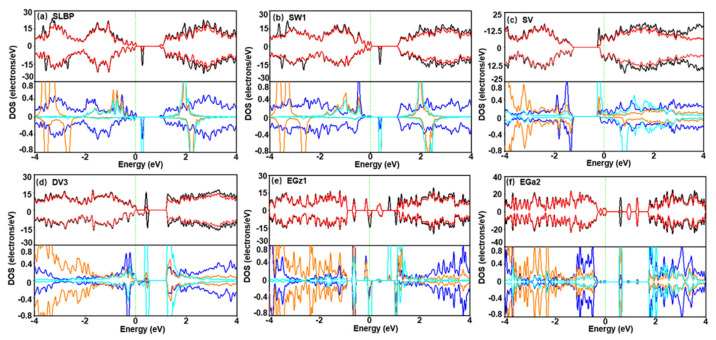
TDOS of adsorption complexes of NO_2_ and SLBP/d−SLBP (black solid line), and PDOS of SLBP/d−SLBP (red solid line) in adsorption complexes (**upper** panel). (**a**) SLBP, (**b**) SW1, (**c**) SV, (**d**) DV3, (**e**) EGz1 and (**f**) EGa2. Partial DOS of p orbitals of N (cyan solid line) and O (orange solid line) atoms, and P atom (blue solid line) (**lower** panel). Green dash line: Fermi level.

**Figure 6 nanomaterials-12-02011-f006:**
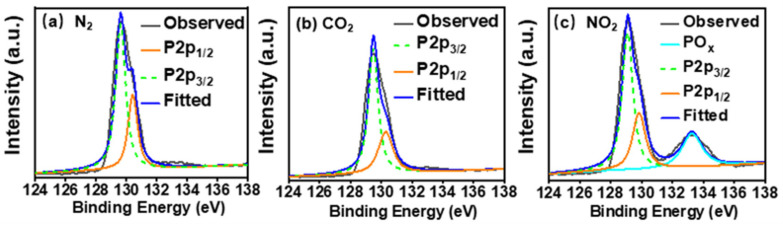
XPS spectra (P 2p) of LBP nanoflakes exposed to N_2_ (**a**), CO_2_ (**b**), or NO_2_ (**c**) for 24h.

**Table 1 nanomaterials-12-02011-t001:** Parameters for the most stable adsorption configurations of gas molecules on perfect SLBP.

Gases	Orientation [a]	*d*_shortest_ [b] (Å)	*E*_ad_ [c] (eV)	*E*_vdW_ (eV)	*E*_vdW_/*E*_ad_ [d]	Δ*q* [e] (e)
NO_2_	**2O*vz***	2.91	−0.225	−0.160	71%	−0.14
SO_2_	** *pz* **	2.91	−0.310	−0.195	63%	−0.11
	**S*vz***	3.07	−0.309	−0.175	57%	−0.09
CO_2_	** *pa* **	3.44	−0.156	−0.124	79%	−0.01
NO	** *pa* **	2.47	−0.200	−0.142	71%	−0.15
CO	** *pa* **	3.37	−0.138	−0.113	82%	−0.01
	** *pa* **	3.41	−0.138	−0.110	80%	−0.02
NH_3_	**N*va***	3.17	−0.207	−0.148	72%	0.04

[a] 2O, S and N indicate two O atoms, one S atom and one N atom pointing toward SLBP; ***v*** and ***p*** indicate that the gas molecule stands vertically or parallel to SLBP; ***z*** and ***a*** represent zigzag and armchair directions. [b] The shortest distance between adsorbed gas molecule and SLBP. [c] *E*_ad_: adsorption energy including vdW correction. A negative value means exothermic adsorption. [d] Ratio of vdW energy (*E*_vdW_) to *E*_ad_. [e] Charge transfer amounts during adsorption. A negative sign “−” represents that a negative charge transfers from SLBP to absorbed gas molecules.

## Data Availability

The data presented in this study are available on request from the corresponding author.

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
