# Peer review of "Unique Interaction between Layered Black Phosphorus and Nitrogen Dioxide"

_nanomaterials, 2022, doi:10.3390/nano12122011_

Round 1

Reviewer 1 Report

The paper “Unique Interaction Between Layered Black Phosphorus and Nitrogen Dioxide” made by Jingjing Zhao, Xuejiao Zhang, Qing Zhao, Xue-Feng Yu, Siyu Zhang and Baoshan Xing present the information about interaction between layered black phosphorus (LBP) and hazardous gases. The authors tried to bring practical arguments to the data obtained by simulation.

 The paper can be published after minor correction. The authors can consider the following:

  1. In Abstract, line 16 exist two “air”
  2. Cytotoxicity data of these nanomaterial should be added or at least mentioned
  3. The authors can insert the number of shells for “2.1. Adsorbent Model” as well as the percentage of surface atoms
  4. Also, at “2.4. Experimental Methods” the authors must insert some data of anaerobic water (at least the COD parameter value)
  5. By sonication the authors obtained the phosphorene or another phase?!
  6. The centrifugation and filtration were carried out under glove box or only dried process?! If all processes, please reformulate the sentences at line 155-158
  7. Lines 158-159: which is “the same flow rate for 3 h”?!
  8. The NO2 or CO2 were pure or under mixture with N2?! Please insert the company name of pure N2
  9. Please redesign Figures 2, 5, 6 because is difficult to observe them even in electronic form
  10. Line 328 “umber”?!
  11. Lines 361-362: “Only a P 2p peak at 129.4 eV is observed” is P 2p1//2 or 2p3//?!
  12. Why the authors not involved another gases as SO2, NH3, etc. in practical determination?!

Author Response

Response to Reviewer :

The paper “Unique Interaction Between Layered Black Phosphorus and Nitrogen Dioxide” made by Jingjing Zhao, Xuejiao Zhang, Qing Zhao, Xue-Feng Yu, Siyu Zhang and Baoshan Xing present the information about interaction between layered black phosphorus (LBP) and hazardous gases. The authors tried to bring practical arguments to the data obtained by simulation. The paper can be published after minor correction. The authors can consider the following:

Response: Thanks for the positive comments. We have revised the manuscript following suggestions to further improve the quality. The revised manuscript is attached. Please see the attachment.

1. In Abstract, line 16 exist two “air”.

Response: Thanks. The sentence was revised. Please see line 16 in the revised manuscript.

2. Cytotoxicity data of these nanomaterial should be added or at least mentioned.

Response: Thanks for the suggestion. Literatures on cytotoxicity of LBP were added into the introduction. Please see lines 55-57 in the revised manuscript.

3. The authors can insert the number of shells for “2.1. Adsorbent Model” as well as the percentage of surface atoms.

Response: Thanks for the suggestion. In this manuscript, a 3 × 1 × 3 supercell containing 36 phosphorus atoms (P1 ‒ P36) was constructed to simulate perfect SLBP. Each P atom of SLBP was covalently bonded to three adjoining P atoms, forming a honeycomb wrinkled structure through sp3 hybridization. This honeycomb wrinkled structure made SLBP containing two atomic layers. The bond length of P-P in the same layer is 2.22 Å, and the bond length of P-P in the upper and lower layers is 2.26 Å. There are 18 atoms in the upper layer (yellow) and 18 atoms in the lower layer (purple), each accounting for 50% of the atoms. Please see lines 107–109 in the revised manuscript and Figure S2 in the revised supporting information.

4. Also, at “2.4. Experimental Methods” the authors must insert some data of anaerobic water (at least the COD parameter value).

Response: Thanks for the suggestion. The water used in experiments was prepared by heating Millipore ultrapure water to 100℃ and degassing with N2 for at least 2h. No COD was included. Please see line 174 in the revised manuscript.

5. By sonication the authors obtained the phosphorene or another phase?

Response: Thanks. LBP was prepared by liquid exfoliation method described in previous study (Angew. Chem. Int. Ed, 2018, 58(2), 467-471). Firstly, the bulk black phosphorus was ground into powders with an agate mortar in the glovebox. The powders were transferred to oxygen-free Millipore ultrapure water and sonicated with a probe for 12 hours. The as-exfoliated suspension was sealed and centrifuged at 10000 rpm for 30 minutes. The supernatant obtained was LBP also called phosphorene. Detailed preparation methods were provided in lines 172-176 of the revised manuscript.

6. The centrifugation and filtration were carried out under glove box or only dried process? If all processes, please reformulate the sentences at line 155-158.

Response: Thanks for the suggestion. The sentences were reformulated. Please see lines 173-178 in the revised manuscript.

7. Lines 158-159: which is “the same flow rate for 3 h”?

Response: Thanks. The injection flow rates of NO2 and CO2 into LBP suspensions were controlled both to be 10 mL/min, in order to make comparable exposure conditions. Please see line 181 in the revised manuscript.

8. The NO2 or CO2 were pure or under mixture with N2? Please insert the company name of pure N2

Response: Thanks for the suggestion. NO2 and N2 mixed gas was purchased from Beijing Lvyuan Dade Biological Tech Co., Ltd. The concentration of NO2 was 200 ppm. High purity CO2 and N2 were obtained from Shenyang Shuntai Special Gas Co., Ltd. The purity of CO2 was >99.5%, and that of N2 was > 99.99%. Please see lines 179-184 in the revised manuscript.

9. Please redesign Figures 2, 5, 6 because is difficult to observe them even in electronic form.

Response: Thanks for the suggestion. The suggestion was adapted. Please see Figures 2, 5, 6 in the revised manuscript.

10. Line 328 “umber”?

Response: Thanks. The word was corrected. Please see line 361 in the revised manuscript.

11. Lines 361-362: “Only a P 2p peak at 129.4 eV is observed” is P 2p1/2 or 2p3/2?

Response: Thanks. The sentence was reformulated. Please see lines 397-399 in the revised manuscript.

12. Why the authors not involved another gases as SO2, NH3, etc. in practical determination?

Response: Thanks for the comments. In this study, the experiments were carried out to prove our findings in theoretical calculation. Through theoretical calculation, we found that the unsaturated defects of LBP and NO2 containing single electron have unique redox interaction. Single electron of both unsaturated LBP and gas molecules were necessary to form the strong interaction. For other gases like CO2 without single electron, the weak physical adsorption is dominated by van der Waals force. Therefore, NO2 and CO2 were selected as representative gases to perform the experiments. Please see lines 390-395 in the revised manuscript.

Reviewer 2 Report

The paper deals with very important topic and  provides mechanism understandings in advance for developing novel nano-technologies for selectively monitoring or treating containing NO2 gas pollutants. This is timely and relevant topic for Nanomaterials journal. The paper is  well-prepared, appropriate methods applied. There are no discussion and conclusions sections in this paper. The literature review section is also missing. The paper needs to be revised, the input of this paper  in the light of other studies in this field need tom be highlighted, discussion and conclusion sections need to be added. 

Author Response

Response to Reviewer :

The paper deals with very important topic and provides mechanism understandings in advance for developing novel nano-technologies for selectively monitoring or treating containing NO2 gas pollutants. This is timely and relevant topic for Nanomaterials journal. The paper is well-prepared, appropriate methods applied. There are no discussion and conclusions sections in this paper. The literature review section is also missing. The paper needs to be revised, the input of this paper in the light of other studies in this field need tom be highlighted, discussion and conclusion sections need to be added.

Response: Thanks for the positive comments and suggestions, which is very valuable for the improvement of our manuscript. The revised manuscript is attached. Please see the attachment. New literatures were added. Please see lines 40-42, 55-57, 66-68 and 90-92 in the revised manuscript. “3. Results and Discussion” section was revised according to the suggestions. Please see lines 241, 306-307, 346 and 397-399 in the revised manuscript. “4. Conclusions” section was revised according to the suggestion. Please see line 419 in the revised manuscript.

Reviewer 3 Report

Nanomaterials

Manuscript ID:           nanomaterials-1709247

Title:                            Unique Interaction Between Layered Black Phosphorus and Nitrogen Dioxide

Two-dimensional nanomaterials exhibit exciting application potentials in air pollution control, among which layered black phosphorus (LBP) has superior performance and is environmental-friendly. In particular, the authors showed for NO that the interaction mechanism is chemisorption on unsaturated LBP defects, while for SO2, NH3, CO2 or CO, the interaction is dominated by van der Waals force (57-82% of the total interaction). Moreover, experiments confirmed that NO2 can oxidize LBP while other gases like CO2 cannot. Finally, the authors provided mechanism in order to understand in advance the under develop novel nanotechnologies. I think that is a very interesting work, well organized and presented; however, some issues should be improved. I recommend publication to Nanomaterials journal after the following major revisions are addresses.

Comment #1

In general, the authors need to improve the introduction section. There are a lot of materials that have been used for the adsorption of acid gases (NO2, SO2) or greenhouse gases (CO2). Please include the following references in the introduction section.

[1].    G.I. Siakavelas, A.G. Georgiadis, N.D. Charisiou, I.V. Yentekakis, M.A. Goula, Cost-effective adsorption of oxidative coupling-derived ethylene using a molecular sieve. Chem. Eng. Technol. 44 (2021) 2041 – 2048.

[2].    A. Pangh, M.D. Esrafili, M.R. Nejad, A DFT investigation of CO and NO adsorption on Cu5Sc and Cu6Sc+ metallic clusters.  Computational and Theoretical Chemistry 1210 (2022) 113657.

[3].    H.V. Ngoc, K.D. Pham, First-principles study on N2, H2, O2, NO, NO2, CO, CO2, and SO2 gas adsorption properties of the Sc2CF2 monolayer. Physica E: Low-dimensional Systems and Nanostructures 141 (2022) 115162.

Comment #2

Which is the innovation and what are the new aspects being introduced on this research topic?

Comment #3

Is it possible for the authors to use TEM or SEM analysis in order to investigate the morphology of this material?

Author Response

Response to Reviewer :

Two-dimensional nanomaterials exhibit exciting application potentials in air pollution control, among which layered black phosphorus (LBP) has superior performance and is environmental-friendly. In particular, the authors showed for NO that the interaction mechanism is chemisorption on unsaturated LBP defects, while for SO2, NH3, CO2 or CO, the interaction is dominated by van der Waals force (57-82% of the total interaction). Moreover, experiments confirmed that NO2 can oxidize LBP while other gases like CO2 cannot. Finally, the authors provided mechanism in order to understand in advance the under develop novel nanotechnologies. I think that is a very interesting work, well organized and presented; however, some issues should be improved. I recommend publication to Nanomaterials journal after the following major revisions are addresses.

Response: Thanks for the positive comments. We have revised the manuscript following suggestions to further improve the quality. The revised manuscript is attached. Please see the attachment.

Comment #1

In general, the authors need to improve the introduction section. There are a lot of materials that have been used for the adsorption of acid gases (NO2, SO2) or greenhouse gases (CO2). Please include the following references in the introduction section.

[1].    G.I. Siakavelas, A.G. Georgiadis, N.D. Charisiou, I.V. Yentekakis, M.A. Goula, Cost-effective adsorption of oxidative coupling-derived ethylene using a molecular sieve. Chem. Eng. Technol. 44 (2021) 2041 – 2048.

[2].    A. Pangh, M.D. Esrafili, M.R. Nejad, A DFT investigation of CO and NO adsorption on Cu5Sc and Cu6Sc+ metallic clusters.  Computational and Theoretical Chemistry 1210 (2022) 113657.

[3].    H.V. Ngoc, K.D. Pham, First-principles study on N2, H2, O2, NO, NO2, CO, CO2, and SOgas adsorption properties of the Sc2CF2 monolayer. Physica E: Low-dimensional Systems and Nanostructures 141 (2022) 115162.

Response: Thanks for the suggestion. The references were added into the introduction. Please see lines 40-42 in the revised manuscript.

Comment #2

Which is the innovation and what are the new aspects being introduced on this research topic?

Response: Thanks. The innovation of this study is unveiling adsorption mechanism of gas pollutants especially NO2 on LBP. This was not solved by previous experimental studies. Contradictions between experimental and computational studies were solved in this manuscript. The findings will benefit environmental engineers to developing novel nanotechnology for treating NO2. To our knowledge, the redox mechanism of NO2 on LBP was firstly clarified.  

Comment #3

Is it possible for the authors to use TEM or SEM analysis in order to investigate the morphology of this material?

Response: Thanks for the suggestion. The suggestion was adopted. Please see Figure S27 and Text S10 in the revised Supporting Information.

Round 2

Reviewer 2 Report

I think authors did solid work and revised they manuscript and improved it. Major comments of reviewers were addressed in revised version of manuscript. Comprehensive answers were provided to reviewers comments. I think with some language polishing which is necessary for scientific paper, this article can be published in current form as I do not have more comments.